# Reproducing complex simulations of economic impacts of climate change with lower-cost emulators

Jun'ya Takakura[1], Shinichiro Fujimori[2], Kiyoshi Takahashi[1], Naota Hanasaki[3], Tomoko Hasegawa[4], Yukiko Hirabayashi[5], Yasushi Honda[6], Toshichika Iizumi[7], Chan Park[8], Makoto Tamura[9], and Yasuaki Hijioka[3]

[1]Social Systems Division, National Institute for Environmental Studies, Tsukuba, 305-8506, Japan
[2]Department Environmental Engineering, Kyoto University, Kyoto, 615-8540, Japan
[3]Center for Climate Change Adaptation, National Institute for Environmental Studies, Tsukuba, 305-8506, Japan
[4]Department of Civil and Environmental Engineering, Ritsumeikan University, Kusatsu, 525-8577, Japan
[5]Department of Civil Engineering, Shibaura Institute of Technology, Tokyo, 135-8548, Japan
[6]Faculty of Health and Sport Sciences, University of Tsukuba, Tsukuba, 305-8577, Japan
[7]Institute for Agro-Environmental Sciences, National Agriculture and Food Research Organization, Tsukuba, 305-8604 Japan
[8]Department of Landscape Architecture, College of Urban Science, University of Seoul, Seoul, 02504, Korea
[9]Global and Local Environment Co-creation Institute, Ibaraki University, Mito, 310-8512, Japan

*Correspondence to*: Jun'ya Takakura (takakura.junya@nies.go.jp)

**Abstract.** Process-based models are powerful tools to simulate the economic impacts of climate change, but they are computationally expensive. In order to project climate-change impacts under various scenarios, produce probabilistic ensembles, conduct on-line coupled simulations, or explore pathways by numerical optimization, the computational and implementation cost of economic impact calculations should be reduced. To do so, in this study, we developed various emulators that mimic the behaviours of simulation models, namely economic models coupled with bio/physical process-based impact models, by statistical regression techniques. Their performance was evaluated for multiple sectors and regions. Among the tested emulators, those composed of artificial neural networks, which can incorporate nonlinearities and interactions between variables, performed better particularly when finer input variables were available. Although simple functional forms were effective for approximating general tendencies, complex emulators are necessary if the focus is regional or sectoral heterogeneity. Since the computational cost of the developed emulators is sufficiently small, they could be used to explore future scenarios related to climate-change policies. The findings of this study could also help researchers design their own emulators under different situations.

## 1 Introduction

Climate change has diverse impacts on society and a wide range of sectors (IPCC, 2014), and these impacts should be quantitatively evaluated to manage overall risks. If we can monetize these impacts, a variety of risks across different sectors

and regions can be considered on a unified scale. This information helps us to design climate-change related policies. It also contributes to estimating the social cost of carbon.

There are a variety of ways to estimate the economic impacts of climate change (Tol, 2002; Stern, 2006; Ciscar et al., 2011; Burke et al., 2015; Takakura et al., 2019). Among the existing approaches, process-based bio/physical impact models coupled with an economic model are widely used and they tend to be elaborate and complex (Weyant, 2017; Diaz and Moore, 2017). Since these process-based simulations can represent underlying bio/physical or economic processes explicitly based on the governing equations, their applications are not limited to prediction of the outcome variables. Process-based simulations can also contribute to deeper understanding of the focal phenomena and they can simulate outcomes under purely counterfactual conditions that never occurred in the past. This cannot be achieved by simpler macroscopic methods (e.g., Burke et al., 2015). Despite these advantages, it is not always easy for researchers to handle these elaborate process-based models (particularly for model users, rather than model developers) because of the model-specific knowledge, skills, and input data that are required. This is especially the case when multiple sectors are targeted because completely different impact models are developed for each sector.

The high computational cost of process-based impact simulations is another problem, and this also makes on-line coupling with other models difficult. On-line coupling of impact models is required, for example, to represent feedback effects of climate-change impacts on climate-change mitigation (Matsumoto, 2019) and many other synergies and trade-offs among sectors (Yokohata et al., 2020). The possibility of simulation under various scenarios or probabilistic ensemble simulation of impacts also depends on the computational cost of the impact simulations. Mainly due to their high computational cost, typically, process-based simulations of the impacts can be conducted under a limited number of scenarios such as Representative Concentration Pathways (RCPs) (van Vuuren et al., 2011). While these scenarios reasonably cover the plausible range of the radiative forcing levels at the end of the 21st century, there are an infinite number of emission pathways which are not included in the discrete RCP scenarios (e.g., intermediate pathway between RCP2.6 and RCP4.5). Recently, particularly after the Paris Agreement, more attention has been paid to the effect of subtler differences in emission pathways (Keywan et al., 2021). When we try to find the optimal pathway by numerical optimization, repetitive calculations of the objective function which we want to minimize or maximize are needed, and if the impacts of climate change are included in the objective function, they also need to be calculated many times until the value of the objective function converges. Ensemble simulation of the impacts is also important to manage the risk because of the probabilistic characteristics of the climate (Mitchell et al., 2017; Mizuta et al., 2017), but this also requires a large number of simulation runs.

Therefore, reducing the implementation and computational costs of impact calculations is useful for many purposes even if representation of the underlying processes is omitted when the focus is on the outcome variable, not on these underlying processes.

One possible way to solve these issues is statistically mimicking the behaviours of the process-based impact simulations. Such approaches are called emulations (Castelletti et al., 2012). In emulations, emulators try to reproduce the relationships

between the inputs and outputs of the impact models regarding the underlying processes as a black box. A simple but widely used way involves expressing the impact by a simple damage function. Such simplification is adopted in several integrated assessment modelling frameworks (Waldhoff et al., 2014; Nordhaus, 2017). The most typical form of such a damage function is a quadratic function (Howard and Sterner, 2017). In this case, the impact of climate change is expressed by a

quadratic function of the mean temperature rise (such simple damage functions are not called emulators in general, but they act in the same way as the so-called emulators). It is also possible for simple damage functions to incorporate socioeconomic conditions. Compared to the simple damage functions, typical climate-change impact emulators adopt relatively complex functional forms. These include multivariate regression or statistical machine learning techniques such as an artificial neural network (Harrison et al., 2013; Oyebamiji et al., 2015; Schnorbus and Cannon, 2014). By using these techniques, emulators

can represent more complex input-output relationships, but existing studies using these techniques mainly focus on bio/physical impacts rather than economic impacts of climate change. In our previous work, it has been demonstrated that the simulated economic impacts of climate change are affected by socioeconomic conditions as well as the climate conditions and there are complex, non-linear interactions (Takakura et al., 2019). Therefore, using such advanced techniques can be beneficial to emulations of the economic impacts of climate change, too.

Besides the choice of functional form, there are multiple options in the selection of the input variables. By leveraging all the information used in the simulation and using sufficiently complex models, it is theoretically possible to perfectly reproduce the results of the simulation by the emulation (Cybenko, 1989). On the other hand, in practical terms, the number of parameters used in the emulation model will increase and it is impossible to identify the parameters based on the limited simulation results. Therefore, we use some representative variables as the input to the emulators by summarizing the original

input data. These input variables should contain information on climate conditions and socioeconomic conditions, and those jointly determine the magnitude of the economic impacts of climate change. What kind of information is important may depend on what kind of impacts we focus on. For example, some impacts can be accurately predicted by changes in temperature, but others may depend more on changes in precipitation or socioeconomic conditions.

To better design emulators, we need to identify important factors which affect performance, i.e., those that determine

how well the emulators can reproduce the results of simulations. However, there have been no systematic comparisons of the attained performance of the emulators considering the above-mentioned factors. The purpose of this study was to develop and evaluate emulators for the projection of the economic impacts of climate change and identify the relationship between the attained performance of emulators and functional forms or input variables. For this purpose, we used the results of economic impact simulations covering many sectors (Takakura et al., 2019). In this study, the results of the original

simulation results were regarded as the 'ground truth', and emulators tried to reproduce the ground truth statistically when corresponding input was given. Various emulators (different functional forms and input variables) were developed and their performance, how well they can reproduce the results of simulations, was systematically compared.

We expect there are two main groups of readers of this article. The first group is those wish to use the emulators developed herein. The second group is the readers who wish to develop their own emulators using their simulation results.

We provide specific information on our development process. This information could be particularly useful for the second group. For the first group, the emulators we have developed can be freely downloaded from a repository (details below) and explored in conjunction with this article to avoid any potential issues in terms of misuse or misinterpretation.

## 2 Materials & Methods

### 2.1 Simulation of the economic impacts of climate change

We used previously published results of simulations, in which up to nine different sectoral economic impacts of climate change were simulated by bio/physical impact models coupled with economic models (Takakura et al., 2019). Here, 'economic models' refers to the methodologies by which bio/physical impacts are monetized regardless of their ways of monetization. We used the simulated economic impacts caused by changes in agricultural productivity (Iizumi et al., 2017; Fujimori et al., 2018), undernourishment (Hasegawa et al., 2016a), heat-related excess mortality (Honda et al., 2014),

cooling/heating demand (Hasegawa et al., 2016b; Park et al., 2018), occupational-health cost (Takakura et al., 2017), hydropower generation capacity (Zhou et al., 2018b), thermal power generation capacity (Zhou et al., 2018a; Zhou et al., 2018c), fluvial flooding (Kinoshita et al., 2018), and coastal inundation (Tamura et al., 2019) due to climate change. In each sector, bio/physical impacts were modelled by specific process-based impact models, and then the impacts were monetized either by multiplying values of statistical life (VSL) (Oecd, 2012) by the damage functions which translate bio/physical

impacts into economic damages (Kinoshita et al., 2018; Tamura et al., 2019) or by a computational general equilibrium (CGE) model (Fujimori et al., 2012; Fujimori et al., 2017). Here, the CGE model is the AIM/Hub model (formerly known as the AIM/CGE model) (Table 1). While the simulations were conducted under a unified climatic and socioeconomic scenario framework and target years, they differ conceptually depending on characteristics of the impacts and the capability of the models. For example, some simulations intend to capture year-by-year fluctuations in impacts, while others focus only on

longer term impacts. Further, sometimes pure process-based models were not used and statistical regression-based methods were also used in hybrid ways. The simulations were conducted sector by sector, and interactions among sectors were not considered. More details on the original process-based economic impact simulations are described in Takakura et al. (2019) and in SI.1.

The simulations were conducted under the Shared Socioeconomic Pathways – Representative Concentration Pathways

(SSP-RCP) scenario matrix (van Vuuren et al., 2013). We used five SSPs (SSP1, SSP2, SSP3, SSP4, and SSP5) and four RCPs (RCP2.6, RCP4.5, RCP6.0, and RCP8.5). Moreover, in order to incorporate the uncertainty in climate projections, we used five different global climate models (GCMs), namely, HadGEM2-ES, IPSL-CM5A-LR, MIROC-ESM-CHEM, GFDL-ESM2M, and NorESM1-M (Hempel et al., 2013). Therefore, there are 100 (5x4x5) scenario runs in total. The computational general equilibrium model covers 17 regions (AIM's 17 regions shown in Table S1), and thus we have economic impacts for

these 17 regions (for sectors whose economic impacts can be simulated for each country, the results were aggregated for the 17 regions).

While it is impossible to evaluate how accurate these simulation results are because of inherent uncertainty in the simulations, we regard these simulation results as the ground truth. We used the results of these simulations to construct and evaluate the emulators.

## 2.2 Overall framework of the emulations

Figure 1 shows the framework of the simulation and the emulation of the economic impacts of climate change. By using the emulators, we want to get results as similar as possible to the results of process-based simulations when the input data or scenario is given. While emulators do not explicitly model the underlying phenomena, they do have parameters, and by tuning these parameters, they can statistically mimic the behaviours (input-output relationship) of simulations.

Here, $y_{s,r,t|sc}$ denotes the simulated economic impact (in %GDP) in sector $s$, in region $r$, in year $t$, under a given scenario $sc$, and $\hat{y}_{s,r,t|sc}$ is the corresponding emulated economic impact. A scenario $sc$ comprises the combination of SSP, RCP, and GCM. The emulated economic impact $\hat{y}_{s,r,t|sc}$ is calculated by the function $f_{s,r}(\cdot)$ receiving the input $\mathbf{x}_{r,t|sc}$ as expressed in equation (1).

$$\hat{y}_{s,r,t|sc} = f_{s,r}\left(\mathbf{x}_{r,t|sc}\right) \tag{1}$$

The emulator (function $f_{s,r}(\cdot)$) is constructed for each sector and region. The input $\mathbf{x}_{r,t|sc}$ is the (vector of) variable(s) which is used to emulate the economic impact in region $r$, year $t$, under a given scenario $sc$. One important characteristic of the input $\mathbf{x}_{r,t|sc}$ is that there is no suffix $s$. This means that the input variable is not sector-specific and common input data can be used across sectors.

## 2.3 Tested emulators

### 2.3.1 Functional forms

We tested a variety of emulators (different functional forms and input variables) ranging from very parsimonious to complex alternatives. For the functional forms, we used ordinary least squares regression (OLS1), ordinary least squares regression with square terms (OLS2), ordinary least squares regression with square and product terms (OLS2i), multi-layer perceptron (MLP), and a recurrent neural network composed of long short-term memory units (LSTM). For the sake of simplicity, we omit the suffixes $s$, $r$, and $sc$ in this section, and the $i$-th variable in vector $\mathbf{x}_t$ is denoted as $x_{t,i}$.

OLS1 is the simplest form of the emulator, expressed as (2).

$$\hat{y}_t = a_0 + \sum_i a_i x_{t,i} \tag{2}$$

OLS2 includes squared terms, and thus can express some curvature in the response.

$$\hat{y}_t = a_0 + \sum_i a_{1i} x_{t,i} + \sum_i a_{2i} x_{t,i}^2 \tag{3}$$

OLS2i has product terms as well as squared terms and it can represent some types of interactions among variables.

$$\hat{y}_t = a_0 + \sum_i a_{1i} x_{t,i} + \sum_i a_{2i} x_{t,i}^2 + \sum_{i \neq j} a_{ij} x_{t,i} x_{t,j} \tag{4}$$

Simple regressions such as these are widely used, but their capability to express complex phenomena is limited. Currently, more elaborate methods based on statistical machine learning techniques such as artificial neural networks (ANNs) are available. Thus, to represent more complex non-linearities and interactions among variables, we also applied ANN-based techniques to the emulations. MLP is a traditional, but effective and widely used, ANN-based technique that can be applied to the purpose of regression, and thus to the emulation. LSTM is also an ANN-based technique designed to handle time-series data and can represent time-dependent characteristics of the data (e.g., cumulative effects in the economic impacts) as well as non-linearities and interactions among variables. Thus, it may better act as the emulator if time-series data are available as the input. While their strict mathematical formulations are lengthy, MLP can be expressed as

$$\hat{y}_t = f(\mathbf{x}_t, \mathbf{W}) \tag{5}$$

where $\mathbf{W}$ is the weights (parameters) of the model. LSTM has time-dependent internal state $\mathbf{s}_t$, and the output and the internal state at time $t$ can be expressed as

$$\hat{y}_t = f(\mathbf{s}_{t-1}, \mathbf{x}_t, \mathbf{W}) \tag{6}$$

$$\mathbf{s}_t = g(\mathbf{s}_{t-1}, \mathbf{x}_t, \mathbf{W}) \tag{7}$$

See, for example, (Goodfellow et al., 2016) for details on MLP and LSTM. Hyperparameters in the ANN-based models were determined based on preliminary examinations. The number of hidden layers and the number of units in each layer were set to 2 and 32, respectively, and the early-stopping technique was used to avoid overfitting of the models.

### 2.3.2 Input variables

When inputting climate conditions into the emulators, the dimension of the data should be reduced. One typical way to do this is to spatially and temporally aggregate the high-resolution original data. For climate data, the most parsimonious choice involves using the global mean temperature, but this method cannot represent regional and seasonal characteristics of climate conditions. Precipitation also plays an important role for some specific sectors (e.g., hydropower generation capacity, fluvial flooding). We prepared several kinds of input data with different spatial and temporal resolutions by aggregating daily gridded near surface temperature and precipitation data generated by GCMs in the Coupled Model Intercomparison Project phase 5 (CMIP5) (Taylor et al., 2011). First, the spatial resolution of the gridded GCM output data was downscaled to 0.5 x 0.5 degrees by bilinear interpolation. We denote this downscaled gridded temperature as $t_{t|sc}(g, d)$ and precipitation as $p_{t|sc}(g, d)$, where $g$ denotes grid and $d$ denotes day of the year. We calculate annual global mean temperature, annual regional mean temperature and precipitation, and quarterly regional mean temperature and precipitation as follows.

$$agt_{t|sc} = \frac{\sum_{g,d} w_g t_{t|sc}(g, d)}{|D_t| \sum_g w_g} \tag{8}$$

$$art_{rs,t|sc} = \frac{\sum_{g \in rs,d} w_g t_{t|sc}(g,d)}{|D_t| \sum_{g \in rs} w_g} \quad (9)$$

$$arp_{rs,t|sc} = \frac{\sum_{g \in rs,d} w_g p_{t|sc}(g,d)}{|D_t| \sum_{g \in rs} w_g} \quad (10)$$

$$qrt_{q,rs,t|sc} = \frac{\sum_{g \in rs,d \in q} w_g t_{t|sc}(g,d)}{|D_{q,t}| \sum_{g \in rs} w_g} \quad (11)$$

$$qrp_{q,rs,t|sc} = \frac{\sum_{g \in rs,d \in q} w_g p_{t|sc}(g,d)}{|D_{q,t}| \sum_{g \in rs} w_g} \quad (12)$$

Here, $|D_t|$ is the number of days in year $t$ and $|D_{q,t}|$ is the number of days belonging to quarter of a year $q$. Quarters are grouped following the calendar year, namely, January-February-March, April-May-June, July-August-September, and October-November-December. Coefficient $w_g$ is a weight which is proportional to the area of the grid $g$. Regions are indicated by the subscript $rs$. Note that $rs$ is based on the classification of SREX's 26 regions defined in (IPCC, 2012) and different from $r$ (Table S2). While our interest is estimating the economic impacts in each of AIM's 17 regions represented

by $r$, each such region contains different climate zones because $r$ is classified from the viewpoint of economic modelling rather than climatic and geographic conditions. Thus, to incorporate heterogeneity in climate conditions within an AIM region, we use $rs$ instead of $r$ to define climate variables.

    For socioeconomic variables, values are based on the SSP scenarios (Kc and Lutz, 2017; Dellink et al., 2017). Based on the population ($pop_{t|sc}(c)$) and GDP ($gdp_{t|sc}(c)$) in country $c$ in year $t$ under a given scenario $sc$, regional population, GDP,

and GDP per capita are calculated as follows. GDP is measured in USD (2005) based on the market exchange rate.

$$pop_{r,t|sc} = \sum_{c \in r} pop_{t|sc}(c) \quad (13)$$

$$gdp_{r,t|sc} = \sum_{c \in r} gdp_{t|sc}(c) \quad (14)$$

$$gpc_{r,t|sc} = gdp_{r,t|sc}/pop_{r,t|sc} \quad (15)$$

    When inputting variables to the emulators, it is desirable that their values be within a limited range to ensure the stability of

210 numerical computation. Effects of biases in GCMs should also be alleviated. For this purpose, we used the relative changes of these variables as inputs to the emulators. For temperature, changes were defined by the difference from the base-period (1991-2010) values. For the other variables, changes were defined by a log ratio to the base-period or base-year (2005) values (Table 2).

**2.4 Comparison**

As explained in section 2.3, we have various types of emulators (functional forms) and candidate input variables. Among the possible combinations, we conducted comparisons under selected practically relevant conditions.

### 2.4.1 Comparison 1

We quantified the performance of the very simple damage functions (OLS1 and OLS2), which only consider the global mean temperature, and compared the performance when regional climate conditions were considered (Table 3). Here, $rs(r)$ represents a set of SREX regions corresponding to an AIM region $r$ (Table S3).

### 2.4.2 Comparison 2

We investigated the effects of considering socioeconomic conditions. It is also expected that there are interactions between climate conditions and socioeconomic conditions. To identify whether such interactions can be expressed by a simple method, we included product terms in OLS2i (Table 4).

### 2.4.3 Comparison 3

In Comparisons 1 and 2, relatively simple functional forms and temporally coarsely aggregated (annual) climate variables are used. Such an aggregation possibly causes loss of information. For example, crop models consider crop calendars and thus the temperature changes in growing and non-growing seasons have different effects on their original simulation results. Regarding the economic impacts, climatic and socioeconomic conditions of the non-target regions can also affect the target region through, for example, trade in the international market, which is simulated by the AIM/Hub model. To investigate these possibilities, seasonal climate variables, climate variables of non-target regions, and socioeconomic variables of non-target regions were included as input variables. Moreover, when the number of input variables becomes large, more complex functional forms may be more suitable. Thus, we tested OLS2 and MLP using these variables (Table 5).

### 2.4.4 Comparison 4

In the previous comparisons, only simultaneous data were used; that is, when emulating the economic impacts in year $t$, climate and socioeconomic conditions in year $t$ are used. Cumulative or carry-over effects can also exist in the simulated impacts. Therefore, including climate and socioeconomic conditions in past years as the input to the emulator can also contribute to better reproduce the results of the economic simulation. To evaluate the effects of inclusion of information in past years, we tested the performance of artificial neural networks which can consider time-series information (LSTM) with time-series data of different length (10-year data to capture relatively short-period effects and 95-year data which can capture the entire simulation period).

## 2.5 Evaluation

### 2.5.1 Evaluation procedure metrics

Parameters in the emulators are optimized based on the simulation results. If, however, we simply optimized these parameters based on the existing data (simulation results) and evaluated them by the same data, the performance of the emulators might be overestimated compared to the situation in which new data are input to the emulators. This phenomenon is known as overfitting or overlearning. To avoid the effects of overfitting, we use the cross-validation strategy. We have simulation results for 100 scenarios (5 SSPs × 4 RCPs × 5 GCMs), and each scenario has 95 (2006-2100) data points. We

divide the 100 scenario results into 4 groups randomly. Three quarters of the data were used to optimize parameters in the emulators (training), and prediction values were obtained for the remaining one quarter of the data (test). This procedure was repeated four times by changing the training and test data, and then we get the results of emulation for all scenarios. That is, four-fold cross-validation is performed.

        In some situations, we want to emulate impacts under scenarios which are drastically different from the scenarios which

are used to develop (or train) the emulators. In order to evaluate the performance of the emulators under such situations, we also conducted cross-validation by GCM and RCP. Cross-validation by GCM means that the emulators are trained by the simulation results of 4 GCMs (5 SSPs × 4 RCPs × 4 GCMs) and tested by the results of the remaining 1 GCM (5 SSPs × 4 RCPs × 1 GCMs). Cross-validation by RCP mean that the emulators are trained by the simulation results of 3 RCPs (5 SSPs × 3 RCPs × 5 GCMs) and tested by the results of the remaining 1 RCP (5 SSPs × 1 RCPs × 5 GCMs).

Optimization of the parameters (training) and prediction (test) of OLS-based emulators were conducted using the lm function in R 3.4.3 (R Core Team, 2017). ANN-based emulators were trained and tested using the Keras library (Chollet, 2015) in Python 3.7.3. The Windows operating system was used in all cases.

### 2.5.2 Evaluation metrics

        The performance of the emulators was evaluated based on the agreement between the results of the simulations and the

emulations. By a chosen emulator, we obtain the values of emulated economic impacts $\hat{y}_{s,r,t|sc}$. We also have the values of the corresponding original simulated economic impact $y_{s,r,t|sc}$. We measured the agreement between $\hat{y}_{s,r,t|sc}$ and $y_{s,r,t|sc}$ by correlation coefficient (r), root mean squared error (RMSE), ratio of RMSE to standard deviation (RSR), and systematic error (bias). These metrics were calculated for each sector and region.

        The computational cost of the emulators was assessed by the number of required input data, the number of parameters

in a model, model object size (memory size required to load a model), prediction time, and training time. This was measured on a PC (CPU: Intel Core i7-8700K (3.70GHz, 6 cores/12 threads), RAM: 32GB, OS: Windows 10 Pro). While the lm function in R was used for OLS-based emulators in the development, they were transplanted to Python for the assessment of computational cost. Thus, both OLS-based emulators and ANN-based emulators were assessed under equal conditions.

**3 Results**

We report results for r in the main text since the three metrics (r, RMSE, and RSR) varied almost parallelly, and systematic errors (biases) were near-negligible for all conditions. Summarized results beyond r (RMSE, RSR, and bias) are available in Tables S4 to S13 of the supplementary material, and individual values for all sectors and regions are available as electronic supplementary material. A higher value of r (i.e., r closer to 1) indicates that results of the emulation are similar to those of the simulation when the biases are negligible. The value of r also indicates how well the variation in the simulation

results is reproduced by the emulation (square of r is equal to the coefficient of determination or the proportion of explained variance).

Figure 2 is the results of Comparison 1. While there is a large variation in the performance of the emulations for individual sectors, the performance for the aggregated economic impacts is relatively good on average even if they only consider global mean temperature rise. This implies that using simple damage functions can be useful to grasp the rough

picture of economic impacts of climate change. On the other hand, when we focus on more minute components, a more elaborate method is required. The effects of including regional climate conditions are distinct in the economic impacts of thermal power generation and fluvial flooding, whose impacts are strongly affected by local precipitation and river flows.

By incorporating socioeconomic variables as inputs to the emulators, there were significant improvements in the performance of the emulations (Fig. 3). The impacts of climate change are determined not only by hazards (climate

conditions), but also by exposure and vulnerability (socioeconomic conditions) (IPCC, 2014). Most current-generation simulations of economic impacts, including the simulations used in this study, take socioeconomic aspects into account. Thus, it is not surprising that emulators could better reproduce the results of simulations by taking socioeconomic variables into account. Note that there is very little improvement in the results with respect to river flooding impacts. This is mainly because the same proportion of the population and GDP distribution data were used in the simulation of the impacts of

fluvial flooding across SSPs due to data availability (Takakura et al., 2019), and the simulated economic impacts (percentage of GDP) were very similar regardless of the socioeconomic conditions.

Inputting more detailed information improves the performance of the emulations. These improvements were more pronounced when more complex functional forms (MLP) were used. The performance of MLP is comparable or worse compared to that of OLS2 when courser input variables were used (leftmost plots in each panel in Fig. 4), whereas MLP

performs better when finer input variables were used (rightmost plots) in most cases. The relative importance of variables differs depending on the modelled sectors. For example, for the agricultural productivity and undernourishment sectors, the inclusion of socioeconomic variables in non-target regions contributed to the improvement in performance. The performance for the fluvial flooding sector jumps when seasonal climate variables and climate variables in non-target regions are used with MLPs. This is probably due to the result of 'leakage' (discussed later).

Consideration of time-series input variables had positive effects for almost all sectors, but it had greatest effects in the hydropower generation sector (the median r improves from 0.48 to 0.78). This was mainly because LSTM could reproduce

the pre-processing of bio/physical impacts simulations before inputting to the economic models. For example, in the simulation of the hydropower generation sector, calculated physical impacts (theoretical hydropower potential) were averaged for every 20 years, and then temporal linear interpolation was applied because this study focused on long-term potential changes due to climate change rather than year-by-year variations (Zhou et al., 2018b). Temporal moving averaging of biological impacts (yields) was also used in the simulations of agricultural productivity and undernourishment. If the original simulations were conducted using these temporally rounded input data, year-by-year input data do not reproduce the original simulation results well. These effects are more obvious when comparing the time-series results of emulation (for example, see Fig. 10), and the results played out just like a low-pass filter was in place.

In general, the more explanatory variables and the more complex functional forms we use, the better the emulators reproduce the results of the simulations. While this tendency is common for all sectors, there are substantial differences in performance between sectors (Fig. 6). This means some sectors' economic impacts are relatively easy to emulate, but others are more difficult even if the complex techniques are used. There were correlations between impact magnitudes and the performance of the emulators (Fig. 7). That means larger impacts tend to be easier to emulate, and consequently, aggregated impacts are also relatively easy to emulate.

As illustrative examples, we explore simulated and emulated results for chosen sectors in "Brazil" (Figs. 8, 9, and 10). The top row in each figure shows the time series of simulated economic impacts for each scenario, and the remaining rows show corresponding emulated economic impacts by different emulators. For aggregated economic impacts, general tendencies could be reproduced even by simple emulators while complex emulators considering socioeconomic conditions could better represent subtle differences among SSPs (Fig. 8). For occupational-health cost sector and hydropower generation sector impacts, obvious differences among SSPs in the simulation results could not be reproduced by simple emulators, but ANN-based complex emulators could reproduce the general tendencies (Figs. 9 and 10). For the hydropower generation sector, even the most complex emulator failed to reproduce some characteristics of the simulation results; that is, the emulator erroneously predicted discernible economic impacts under SSP1 and SSP4.

The performance of emulation can vary depending on how the training and test data are chosen. The results shown above are based on cross-validation with randomly selected scenarios for training and testing. Figure 11 shows the comparison of the performance between different cross-validation procedures for the aggregated impacts as an example. Here, the performance is shown by the RMSE normalized by the pooled standard deviation (RSR), not by the correlation coefficient, because the standard deviation of each test data formulation, which affects the value of the correlation coefficient, differs across the selected GCMs or RCPs. Summarized results for each sector and indices beyond RSR (r, RMSE, and bias) are available in Tables S14 to S33. When the emulators were trained excluding the results of RCP8.5 (the highest emission pathway), and then tested by the results of RCP8.5 (RCP8.5 left condition), the performance was apparently worse compared to the other conditions. Except for RCP8.5 left condition, the performance was reasonably similar across conditions when simpler models and input data are used. If complex models and finer input variables were used, the performance was worse when cross-validated by GCM or cross-validated by RCP compared to the random cross-validation.

The computational cost of the developed emulators was sufficiently small in the prediction phase, while training requires some time for ANN-based emulators. Table 7 shows the computational cost for selected conditions. Even if the most complex emulators are used, they require only 723 (679 to 1347) milliseconds for the calculation of the economic impact for a century. From the viewpoint of computation time required for the prediction, both the OLS-based and ANN-based models can meet the requirement of the emulators. However, it should be noted that the time required to prepare the input variables is not included in this assessment and it depends on the situations.

## 4 Discussion

In this study, we developed various kinds of emulators and systematically evaluated their performance. We explored differences in emulator performance among sectors and the relationship between model complexity and performance. The aggregated economic impact was relatively easily emulated even by simple emulators with limited input variables. The dominant contributors of aggregated impact were the heat-related excess mortality and occupational-health cost sectors (Takakura et al., 2019) as also shown in the SI 2, and the economic impacts of these two sectors were also relatively easily emulated. There were clear relationships between temperature rise and the simulated impacts in these two sectors (Honda et al., 2014; Takakura et al., 2017), and almost all regions were impacted in the same direction. Moreover, where impacts were large, emulator performance tended to be better as shown in Fig. 7. Temperature-dependent impacts tend to be large and easy to emulate, while precipitation-dependent impacts tend to be small and difficult to emulate. Although it is not clear whether this correlation reflects a causal relationship or is just a coincidence, these characteristics contributed to the higher performance of emulations of aggregated impacts particularly when simple functional forms were used. If we only focus on the aggregated economic impacts of climate change, a simple damage function which only leverages global mean temperature is worth using provided that we regard the original simulation results as valid. On the other hand, some sectors' and regions' impacts were difficult to emulate by simple emulators and consideration of more input variables and more complex functional forms could improve the performance. Therefore, if we focus on sectoral or regional issues (e.g., inequality among regions or sectors), conventional simple damage functions may not be adequate tools and ANN-based or other complex techniques may be necessary.

For the agricultural productivity and undernourishment sectors, the performance of the emulations was low unless socioeconomic conditions of non-target regions were incorporated. Since comparative advantages (or disadvantages) in the international food market and global food demands play important roles in simulations of the impacts in these sectors, it is reasonable that non-target regional information contributed to improve the performance of the emulations. Such beyond-the-border effects have not been considered in previous studies using damage functions or emulators, but our results shed light on the importance of this factor. It is also noteworthy that these improvements were more distinct when MLPs, which can represent complex interactions among variables, were used as the emulators.

In terms of the results for the fluvial flooding sector, inclusion of non-target regions' quarterly climate variables with MLP caused a drastic jump in the performance of the emulators. This is puzzling because in the simulation of the impacts of

fluvial flooding, effects of international trading are not considered explicitly (Kinoshita et al., 2018). We suspect this is caused by the leakage because of the characteristics of the simulation data used in this study. In the field of statistical machine learning, the word leakage means that models have access to some information on the characteristics of the test dataset even if the test and training datasets are separated (Kaufman et al., 2011). In this study, we separated the dataset into training and test datasets depending on the scenarios. When a certain scenario (for example, SSP1-RCP2.6-HadGEM2-ES) is used in the test dataset, it is not included in the training dataset. By doing this, we can evaluate how the trained emulators will work when a new unknown scenario is given. However, in the case of fluvial flooding, the simulated impacts expressed by percentage of GDP are very similar among SSPs (Takakura et al., 2019). For example, the simulated impacts (%GDP) in SSP1-RCP2.6-HadGEM2-ES are almost identical to those in SSP2-RCP2.6-HadGEM2-ES, SSP3-RCP2.6-HadGEM2-ES, SSP4-RCP2.6-HadGEM2-ES, and SSP5-RCP2.6-HadGEM2-ES, and some of these datasets are included in the training dataset. In such a situation, overfitting can result in apparently high performance in the cross-validation even if its actual ability for a new input dataset is low. Therefore, apparently high performance in the fluvial flood sector should be interpreted with caution.

In the sectors of hydropower generation and thermal power generation, even using the complex emulators with finer input variables, the attained performance remained relatively low. This implies required information to reproduce the simulation results is missing from the input data. In the AIM/Hub model, there are SSP-dependent assumptions other than population and GDP, particularly related to energy policies (Fujimori et al., 2017). These policies depend on the narratives of the SSP storylines, not just quantitative socioeconomic information such as population or GDP. In addition, in the AIM/Hub model, adoption of power generation technology is decided by a discrete choice model (Fujimori et al., 2014). Thus, the degree of reliance on a certain kind of power generation can also be discrete or non-continuous depending on the SSP-dependent assumptions in the AIM/Hub model. For example, a certain region does not rely on the hydropower generation at all in some situations, but once the hydropower generation technology becomes economically competitive compared to other power generation technologies, hydropower generation plants will be installed in the model. In the former situation, changes in the hydropower generation capacity do not affect the economy at all, but do in the latter situation. This difference cannot be predicted by the emulators, since they cannot be represented only by climate conditions, GDP, and population.

To improve the performance of the economic impact emulations, should we construct more complex emulators and consider more information? For example, in power generation sectors, model-specific assumptions regarding the energy system could be used as additional input variables and this might improve the emulation performance. If a sufficient number of simulation results are available, this strategy may work. An alternative approach is refraining from reproducing the complex behaviour of the energy system in the simulation model by an emulator, and partly using the original simulation model. For example, in simulations of the impacts of hydropower generation and thermal power generation, bio/physical impacts (theoretical hydropower potential and river flow) are simulated by a global hydrological model, whose computational cost is high (around 15 to 20 hours for one scenario), while the economic impacts are simulated by an economic model, whose computational cost is relatively low (around 1.5 hours for one scenario) compared to that of the

hydrological model. Therefore, if we can only emulate bio/physical impacts, the computational cost of economic impact estimation can be reduced even if we use the original simulation model for the economic part. Such kind of model separations will become important particularly if we focus, for example, on interactions among different sectors (Harrison et al., 2016). Another possibility is constructing SSP-specific emulators. In this study, since we aimed to explore new socioeconomic pathways (e.g., intermediate pathway between SSP1 and SSP2) one common emulator was constructed for different socioeconomic pathways. On the other hand, if we fix the socioeconomic pathways to consider, it is possible to incorporate SSP-specific assumptions into the emulators by separating the models by SSPs. This option could be pursued depending on the purpose of the studies.

Even without introducing overly complex models or considering excessively specific information, there are several techniques which may improve the performance of emulation. For example, variable selection is a widely used technique pursuant of developing parsimonious models and avoiding overfitting. We tested the simple step-wise variable selection based on Akaike's information criterion, which can easily be applied to an OLS-based technique and the results are shown in SI 3. Optimization of the hyperparameters, e.g., the number of units the number of hidden layers, the batch size for training in ANN, can also be effective. In addition to optimizing or modifying the models used in this study, other kind of models such as support vector regression, random forest regression, k-nearest neighbours regression, etc., may also be effective. If we adopt techniques like Gaussian process regression, uncertainty of the predicted value can also be assessed. While we did not investigate these techniques in this study, this represents an important direction for future research.

While there is substantial room for improvement, the emulators developed in this study can be used as tools to explore various other future scenarios with limited computational and implementation cost. Technically, applying ANN-based techniques to economic impact emulation is one of the novelties of this study, and we demonstrated that these techniques can improve the performance of the emulations. However, we do not claim researchers should always use ANN-based (or similar statistically complex) techniques in economic impact emulations. There is a non-negligible trade-off between model complexity and performance. While computational cost of emulation is small in the calculation (prediction) phase as shown in Table 7, even by the most complex emulator used in this study, the availability of input variables is context-specific. For example, the cost of preparing or generating sub-yearly regional climate variables should also be considered. We disclose the source code for the OLS-based and ANN-based emulators developed herein. Sector-specific skills and knowledge are not necessarily needed to use this code and thus the implementation cost is much smaller than that of the original simulation models, particularly if the pre-trained models are used. Nevertheless, transplanting the ANN-based emulators to other modelling languages, if necessary, is not always a trivial task, because of the required software libraries. On the other hand, it is much easier to transplant OLS-based emulators in any modelling language because they only require arithmetic multiplication and addition.

While sector-specific skills and knowledge are not always necessary to use the developed emulators, users should be aware of the statistical context of the emulators and evaluation results. Firstly, cross-validation is a powerful tool to evaluate the emulators' performance without the influence of overfitting, and we can rely on the results of cross-validation to choose

adequate models in most cases. However, leakage can pass the cross-validation test unlike simple overfitting. While there is no perfect solution to detect the existence of leakage, it can be effective to think about the actual situation in which the developed emulators will be used. For example, if the emulators will be used to estimate economic impacts under different RCPs or substantially different emission pathways, which are not included in the training data, cross-validation by RCPs can be effective to estimate the actual performance of the emulators in that situation. Suspected leakage shown in Fig. 4 can be detected by this strategy (SI 4). Secondly, regression models are not good at extrapolation, as shown in RCP8.5 left condition in Fig. 11. Fortunately, RCP8.5 left condition is a hypothetical situation only for the purpose of the evaluation. In the actual situations in which the emulators will be used, the simulation results under RCP8.5 are included in the training data and emission pathways higher than RCP8.5 are nearly impossible considering the current world situation (Hausfather and Peters, 2020). Therefore, as for the emission pathway, the problem of extrapolation will not be a serious issue in practical terms, but we should be aware whether an emulated scenario is inside or outside the range of the original simulations. Thirdly, overfitting should be avoided as a general rule, but some sort of overfitting can be allowed depending on the purpose of the studies. For example, constructing SSP-specific emulators is possible as discussed above. These emulators overfit to the corresponding SSPs (socioeconomic pathways) and thus will not work well under different socioeconomic pathways. On the other hand, as long as the purpose of the emulation is to explore future scenarios other than socioeconomic pathways, overfitting to the SSPs will not be a problem. Such judgements may be difficult. In general, however, unwanted and unexpected characteristics of statistical models tend to emerge when more complex models are used. Therefore, it is conservative, but can be safer, to choose a simpler model if it meets the requirement of the studies when the users are not confident about the model characteristics.

Both simple and complex emulators have advantages and disadvantages. We cannot conclude which emulator is the "best" one, because it depends on the purpose and situation, but we can give a general guideline to choose a suitable emulator based on the results of this study. This is that simple emulators are effective for approximating global general tendencies, but complex emulators are necessary if the focus is regional or sectoral heterogeneity. Through a systematic comparison of different emulators, ranging from very parsimonious through to complex alternatives, the findings of this study can help researchers choose and implement the most suitable emulators for their purposes and situations.

## 5 Limitations and future study

We used the results of simulations as the ground truth, and the emulators were optimized to reproduce the results of the simulations. While we used state-of-the-art simulation results (Takakura et al., 2019), the simulations themselves contain uncertainties and inaccuracies. Thus, even if the emulators could reproduce the results of the simulations, this does not necessarily mean the economic impacts estimated by the emulators are accurate.

In this study, we assumed that climate data from GCMs were given and representative climate variables could be calculated by aggregating (or upscaling) them. On the other hand, particularly when the emulator is used as a component of a typical integrated assessment model, climate data are calculated by a simple climate model and only the global mean

temperature is available. In such a situation, downscaling is necessary if the emulator requires regional or seasonal climate variables. This is possible using, for example, a pattern scaling technique (Herger et al., 2015; Osborn et al., 2016). In this case, the overall performance of the emulation should be evaluated including this pre-processing.

We should also consider the division of roles between models. In this study, the emulators played the roles of both bio/physical impact models and economic models. As discussed above, it can be difficult to incorporate the assumptions

used in the economic models into the emulators. To avoid such difficulties, it may be better for emulators to focus on bio/physical impacts, with the economic impacts being calculated by an appropriate economic model. In general, the computational costs of economic models are lower compared to those of bio/physical impact models, and thus, it is prudent to consider this option. It may also be desirable from the viewpoint of representing interactions among sectors, but more investigations, model developments, and validations are needed to model complex interactions. The optimal configuration of

the model cascade should be decided considering prediction accuracy, computational cost, and the purpose of studies.

**Code and data availability**

Code and data to reproduce the results in this paper are available at http://doi.org/10.5281/zenodo.4692496. There are two directories in the repository. The directory named RTU contains Ready-To-Use code and sample data for users of the developed emulators. The directory named REPRODUCTION contains code and data that are required to reproduce the

490 results reported in this paper.

**Author contributions**

J.T. conducted the formal analysis and prepared the original draft. J.T., S.F., K.T., N.H., T.H., Y. Hirabayashi, Y. Honda, T.I., C.P., and M.T. provided the resources. Y. Hijioka supervised the study. All authors reviewed and/or edited the manuscript.

**Competing interests**

Until Feb. 2016, J.T. was employed by Toshiba Corporation, whose business is related to hydro and thermal power generation. The other authors declare no competing interests.

**Acknowledgement**

This research was supported by the Environment Research and Technology Development Fund (JPMEERF15S11400 and

500 JPMEERF20202002) of the Environmental Restoration and Conservation Agency of Japan.

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

**Table 1: List of modelled sectors. In principle, the results of simulations obtained in Takakura et al. (2019) were used.**
*Definition of the baseline (no-climate change condition) was changed slightly compared to that of the original study.
#Original results imputed by emulation-like technique due to data unavailability.

| Simulated economic impact | Way of monetization |
| --- | --- |
| Agricultural productivity* | CGE model |
| Undernourishment | CGE model + VSL |
| Heat-related excess mortality | VSL |
| Cooling/heating demand | CGE model |
| Occupational-health cost | CGE model |
| Hydropower generation capacity | CGE model |
| Thermal power generation capacity | CGE model |
| Fluvial flooding | Economic damage function |
| Coastal inundation# | Economic damage function |

**Table 2: Candidate input variables. Socioeconomic variables are defined for AIM's 17 regions, while climatic variables are defined for SREX's 26 regions.**

| Variable name | Variable | Spatial resolution | Temporal resolution |
| --- | --- | --- | --- |
| $\Delta agt_{t|sc}$ | Temperature | Global | Annual |
| $\Delta art_{rs,t|sc}$ | Temperature | SREX 26 regions | Annual |
| $\Delta arp_{rs,t|sc}$ | Precipitation | SREX 26 regions | Annual |
| $\Delta qrt_{q,rs,t|sc}$ | Temperature | SREX 26 regions | Quarterly |
| $\Delta qrp_{q,rs,t|sc}$ | Precipitation | SREX 26 regions | Quarterly |
| $\Delta pop_{r,t|sc}$ | Population | AIM 17 regions | Annual |
| $\Delta gdp_{r,t|sc}$ | GDP | AIM 17 regions | Annual |
| $\Delta gpc_{r,t|sc}$ | GDP per capita | AIM 17 regions | Annual |

**Table 3: Models and input variables in Comparison 1. Input variables are used to emulate the economic impact in year $t_1$ in region $r_1$ for each sector.**

| Emulator | Input variables | |
| --- | --- | --- |
| OLS1/OLS2 | $\mathbf{x}_{r,t|sc} = agt_{t|sc}$ | $t = t_1$ |
| OLS1/OLS2 | $\mathbf{x}_{r,t|sc} = (\{art_{rs,t|sc}\}, \{arp_{rs,t|sc}\})$ | $t = t_1, rs \in rs(r_1)$ |

**Table 4: Models and input variables in Comparison 2. Input variables are variables used to emulate the economic impact in year $t_1$ in region $r_1$ for each sector.**

| Emulator | Input variables | |
|---|---|---|
| OLS2/OLS2i | $\mathbf{x}_{r_1,t_1\|sc} = (\{art_{rs,t\|sc}\}, \{arp_{rs,t\|sc}\})$ | $t = t_1, rs \in rs(r_1)$ |
| OLS2/OLS2i | $\mathbf{x}_{r_1,t_1\|sc} = (\{art_{rs,t\|sc}\}, \{arp_{rs,t\|sc}\}, \{pop_{r,t\|sc}\}, \{gdp_{r,t\|sc}\}, \{gpc_{r,t\|sc}\})$ | $t = t_1, rs \in rs(r_1), r = r_1$ |

**Table 5: Models and input variables in Comparison 3. Input variables are variables used to emulate the economic impact in year $t_1$ in region $r_1$ for each sector.**

| Emulator | Input variables | |
|---|---|---|
| OLS2/MLP | $\mathbf{x}_{r_1,t_1\|sc} = (\{art_{rs,t\|sc}\}, \{arp_{rs,t\|sc}\}, \{pop_{r,t\|sc}\}, \{gdp_{r,t\|sc}\}, \{gpc_{r,t\|sc}\})$ | $t = t_1, rs \in rs(r_1), r = r_1$ |
| OLS2/MLP | $\mathbf{x}_{r_1,t_1\|sc} = (\{art_{rs,t\|sc}\}, \{arp_{rs,t\|sc}\}, \{pop_{r,t\|sc}\}, \{gdp_{r,t\|sc}\}, \{gpc_{r,t\|sc}\})$ | $t = t_1, rs \in rs(r_1), \forall\, r$ |
| OLS2/MLP | $\mathbf{x}_{r_1,t_1\|sc} = (\{qrt_{q,rs,t\|sc}\}, \{qrp_{q,rs,t\|sc}\}, \{pop_{r,t\|sc}\}, \{gdp_{r,t\|sc}\}, \{gpc_{r,t\|sc}\})$ | $t = t_1, rs \in rs(r_1), r = r_1, \forall\, q$ |
| OLS2/MLP | $\mathbf{x}_{r_1,t_1\|sc} = (\{qrt_{q,rs,t\|sc}\}, \{qrp_{q,rs,t\|sc}\}, \{pop_{r,t\|sc}\}, \{gdp_{r,t\|sc}\}, \{gpc_{r,t\|sc}\})$ | $t = t_1, rs \in rs(r_1), \forall\, r, \forall\, q$ |
| OLS2/MLP | $\mathbf{x}_{r_1,t_1\|sc} = (\{art_{rs,t\|sc}\}, \{arp_{rs,t\|sc}\}, \{pop_{r,t\|sc}\}, \{gdp_{r,t\|sc}\}, \{gpc_{r,t\|sc}\})$ | $t = t_1, \forall rs, r = r_1$ |
| OLS2/MLP | $\mathbf{x}_{r_1,t_1\|sc} = (\{art_{rs,t\|sc}\}, \{arp_{rs,t\|sc}\}, \{pop_{r,t\|sc}\}, \{gdp_{r,t\|sc}\}, \{gpc_{r,t\|sc}\})$ | $t = t_1, \forall rs, \forall r$ |
| OLS2/MLP | $\mathbf{x}_{r_1,t_1\|sc} = (\{qrt_{q,rs,t\|sc}\}, \{qrp_{q,rs,t\|sc}\}, \{pop_{r,t\|sc}\}, \{gdp_{r,t\|sc}\}, \{gpc_{r,t\|sc}\})$ | $t = t_1, \forall rs, r = r_1, \forall q$ |
| OLS2/MLP | $\mathbf{x}_{r_1,t_1\|sc} = (\{qrt_{q,rs,t\|sc}\}, \{qrp_{q,rs,t\|sc}\}, \{pop_{r,t\|sc}\}, \{gdp_{r,t\|sc}\}, \{gpc_{r,t\|sc}\})$ | $t = t_1, \forall rs, \forall r, \forall q$ |

**Table 6: Models and input variables in Comparison 4. Input variables are used to emulate the economic impact in year $t_1$ in region $r_1$ for each sector.**

| Emulator | Input variables | |
|---|---|---|
| MLP | $\mathbf{x}_{r_1,t_1\|sc} = (\{qrt_{q,rs,t\|sc}\}, \{qrp_{q,rs,t\|sc}\}, \{pop_{r,t\|sc}\}, \{gdp_{r,t\|sc}\}, \{gpc_{r,t\|sc}\})$ | $t = t_1, \forall rs, \forall r, \forall q$ |
| LSTM | $\mathbf{x}_{r_1,t_1\|sc} = (\{qrt_{q,rs,t\|sc}\}, \{qrp_{q,rs,t\|sc}\}, \{pop_{r,t\|sc}\}, \{gdp_{r,t\|sc}\}, \{gpc_{r,t\|sc}\})$ | $t = t_1, \cdots, t_1 - 9, \forall rs, \forall r, \forall q$ |
| LSTM | $\mathbf{x}_{r_1,t_1\|sc} = (\{qrt_{q,rs,t\|sc}\}, \{qrp_{q,rs,t\|sc}\}, \{pop_{r,t\|sc}\}, \{gdp_{r,t\|sc}\}, \{gpc_{r,t\|sc}\})$ | $t = t_1, \cdots, t_1 - 94, \forall rs, \forall r, \forall q$ |

**Table 7: Computational cost of the developed emulators. Median (minimum, maximum) of 17 regional, 9 sectoral and aggregated impacts results are shown. OLS-based models and ANN-based models were implemented by statsmodels library and keras library in Python respectively. OLS2 (RMT+P+S): OLS2 with regional mean temperature precipitation and socioeconomic variables. MLP (RMT+P+S): MLP with regional mean temperature precipitation and socioeconomic variables. MLP (All variables): MLP with all the input variables. LSTM (All variables, 95): LSTM with all the input variables for 95 years.**

| Model (input) | Number of input variables | Number of parameters | Model object size (MB) | Prediction time (second/scenario) | Training time (second) |
|---|---|---|---|---|---|
| OLS2 (GMT) | 1 (1, 1) | 3 (3, 3) | 1.470 ( 1.470, 1.470) | 0.002 (0.002, 0.004) | 0.010 (0.008, 0.018) |
| OLS2 (RMT+P) | 6 (2, 10) | 13 (5, 21) | 5.291 ( 2.234, 8.348) | 0.003 (0.002, 0.015) | 0.018 (0.010, 0.029) |
| OLS2 (RMT+P+S) | 9 (5, 13) | 19 (11, 27) | 7.583 ( 4.527, 10.642) | 0.004 (0.003, 0.006) | 0.022 (0.014, 0.034) |
| MLP(RMT+P+S) | 9 (5, 13) | 1409 (1281, 1537) | 7.711 ( 7.710, 7.711) | 0.054 (0.050, 0.133) | 3.601 (1.024, 13.349) |
| MLP (All variables) | 259 (259, 259) | 9409 (9409, 9409) | 7.711 ( 7.711, 7.711) | 0.134 (0.129, 0.224) | 6.552 (1.375, 15.350) |
| LSTM (All variables, 95) | 24605 (24605, 24605) | 45729 (45729, 45729) | 96.461 (96.454, 96.461) | 0.723 (0.679, 1.347) | 689.066 (153.160, 1761.409) |

660

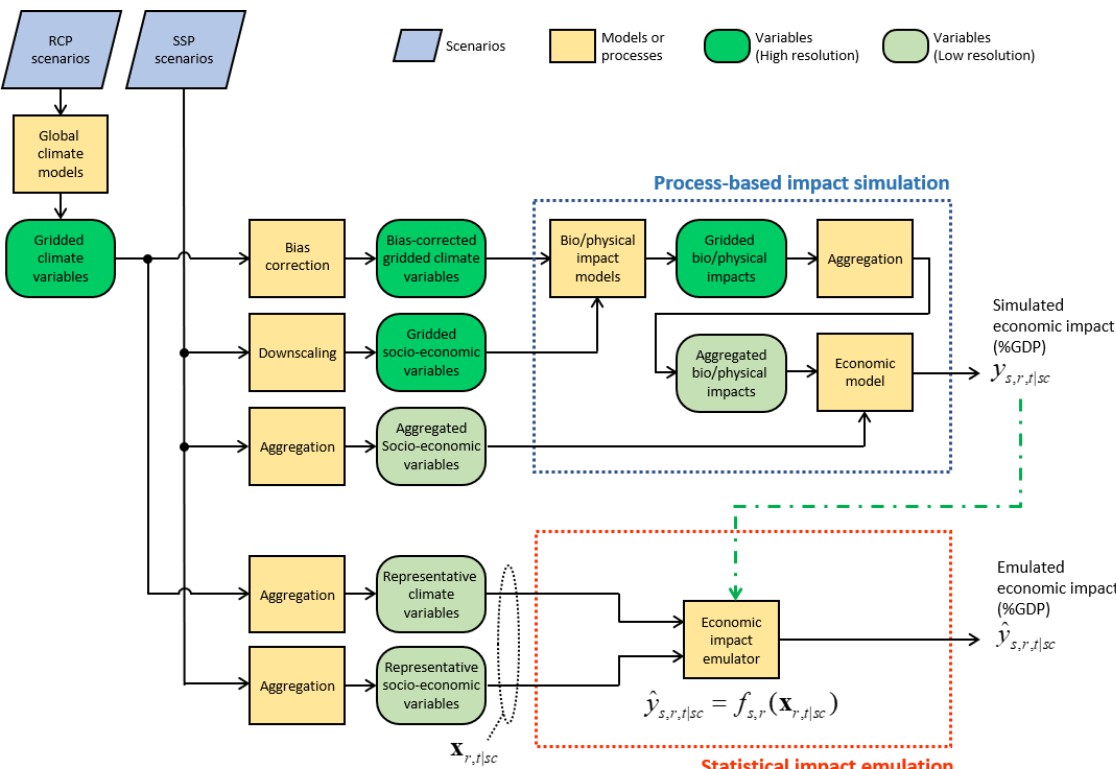

Figure 1: Overall framework of simulation and emulation of economic impacts of climate change. Simulated and emulated economic impacts in sector $s$, region $r$, year $t$ under a scenario $sc$ are denoted as $y_{s,r,t|sc}$ and $\hat{y}_{s,r,t|sc}$, respectively. The parameters of the emulator are determined based on the simulated economic impact (represented by the green dash-dot-dash arrow).

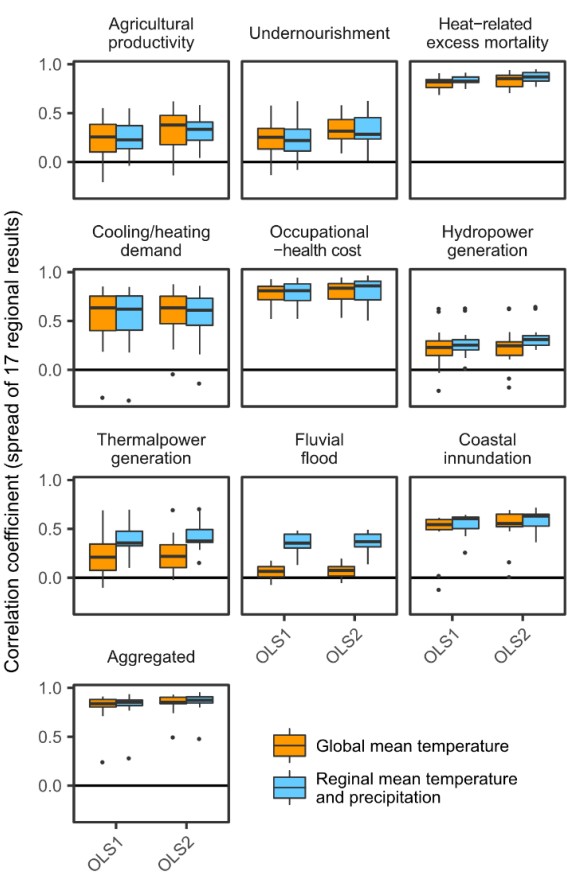

**670**

**Figure 2: Performance of emulations in Comparison 1. Correlation coefficients between simulation results and emulation results are shown. Bars and edges of boxes represent medians and first/third quantile values among 17 regional results. The ends of the whiskers show the minimum and maximum values, while outliers are denoted by dots if they exist.**

**675**

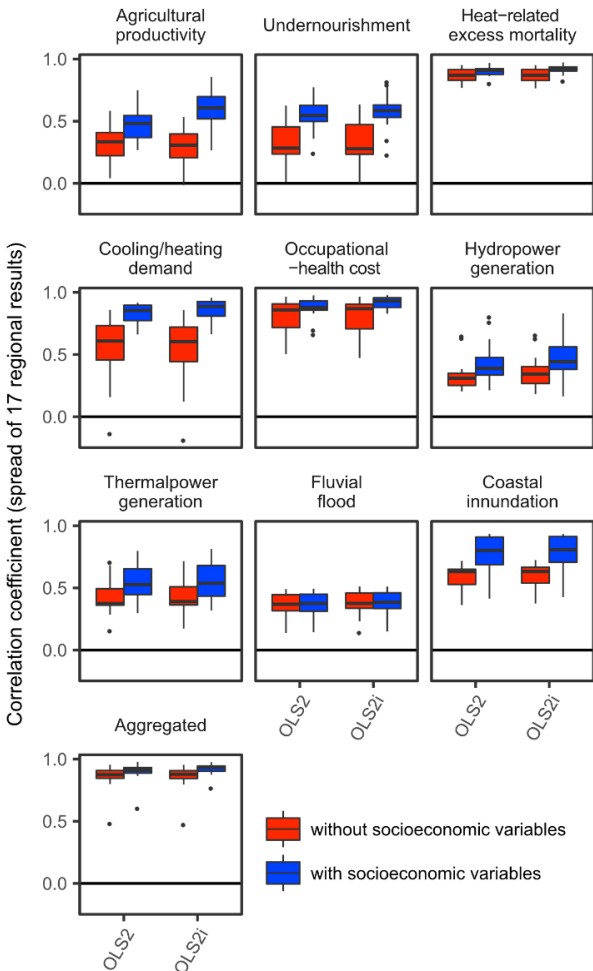

**Figure 3: Performance of emulations in Comparison 2. Correlation coefficients between simulation results and emulation results are shown.**

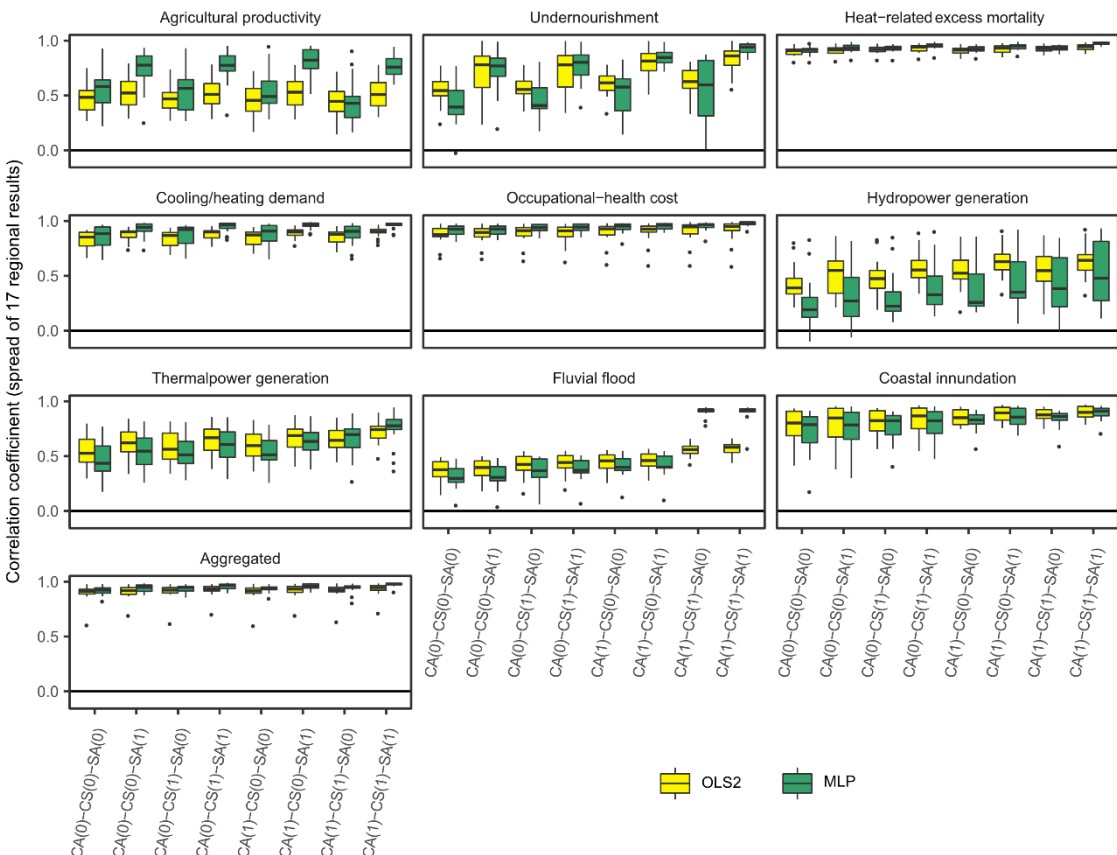

**Figure 4: Performance of emulations in Comparison 3.** Correlation coefficients between simulation results and emulation results are shown. CA(1) denotes that climate variables for all regions (including non-target region) are used. CS(1) denotes that seasonal climate variables are used. SA(1) denotes that socioeconomic variables for all regions (including non-target region) are used.

680

685

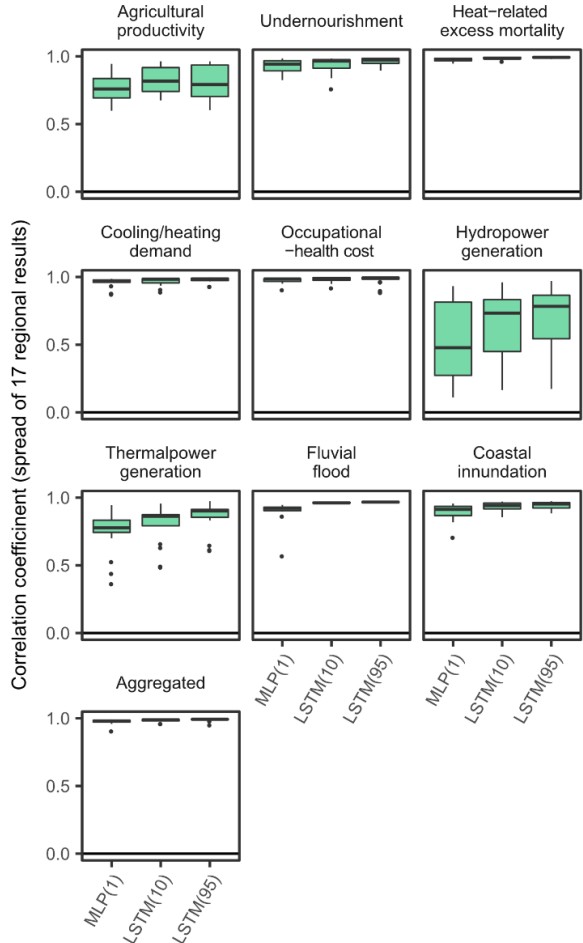

**Figure 5: Performance of emulations in Comparison 4. Correlation coefficients between simulation results and emulation results are shown. MLP(1) denotes that MLPs are used as the emulators and only climatic and socioeconomic variables for the target year are used. LSTM(10) and LSTM(95) denote that LSTMs are used as the emulators and the climatic and socioeconomic variables for 10 and 95 years are used, respectively.**

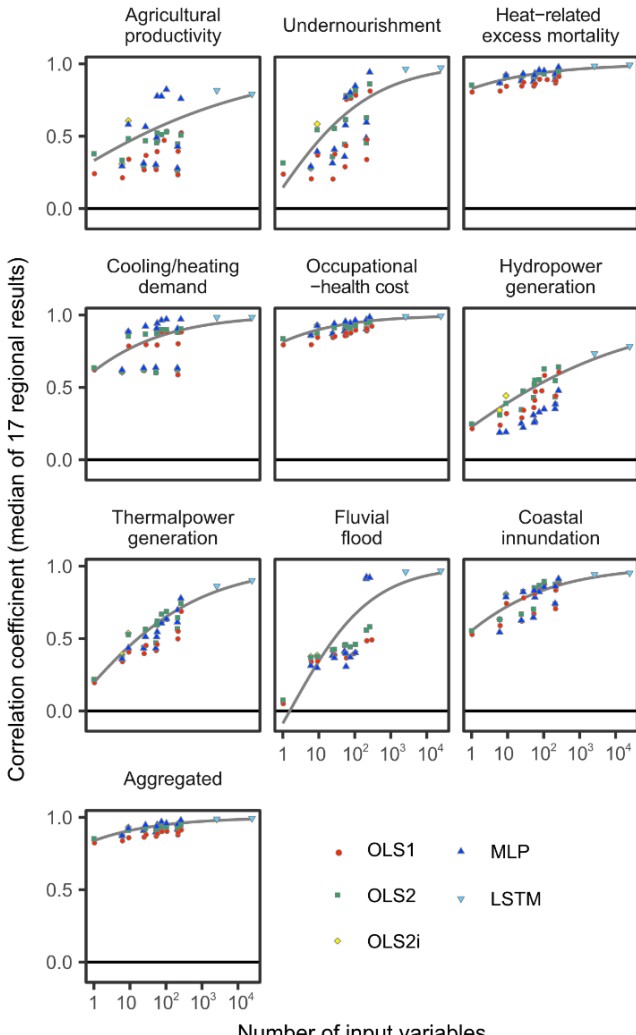

**Figure 6: Relationship between number of input variables and performance of emulators. Medians of 17 regional results are plotted as points. Fitted curves are produced for the frontiers (Pareto optimal corresponding to each number of input variables) by beta regression. When time-series input data are used, the number of input variables is multiplied by the length of the time series.**

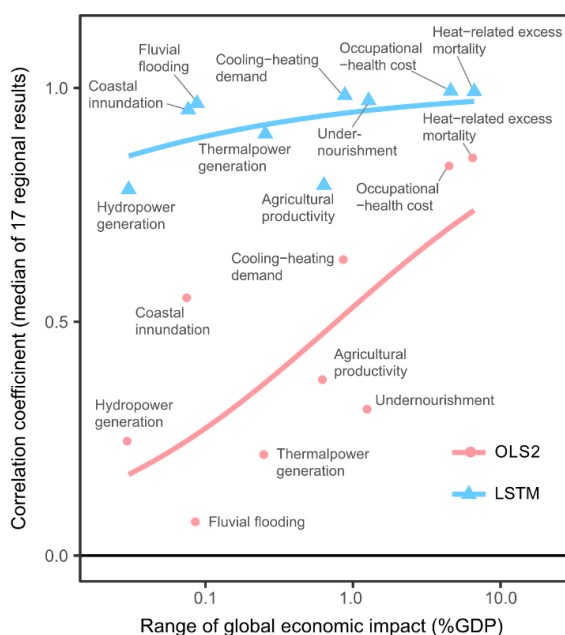

**Figure 7: Relationship between range of economic impacts (global) and performance of emulators. Each point represents a sector, and the median of 17 regional results is plotted as the y-axis value. Fitted curves are produced by beta regression. OLS2 uses only the global mean temperature as the input variable, and LSTM uses all the prepared input variables for 95 years.**

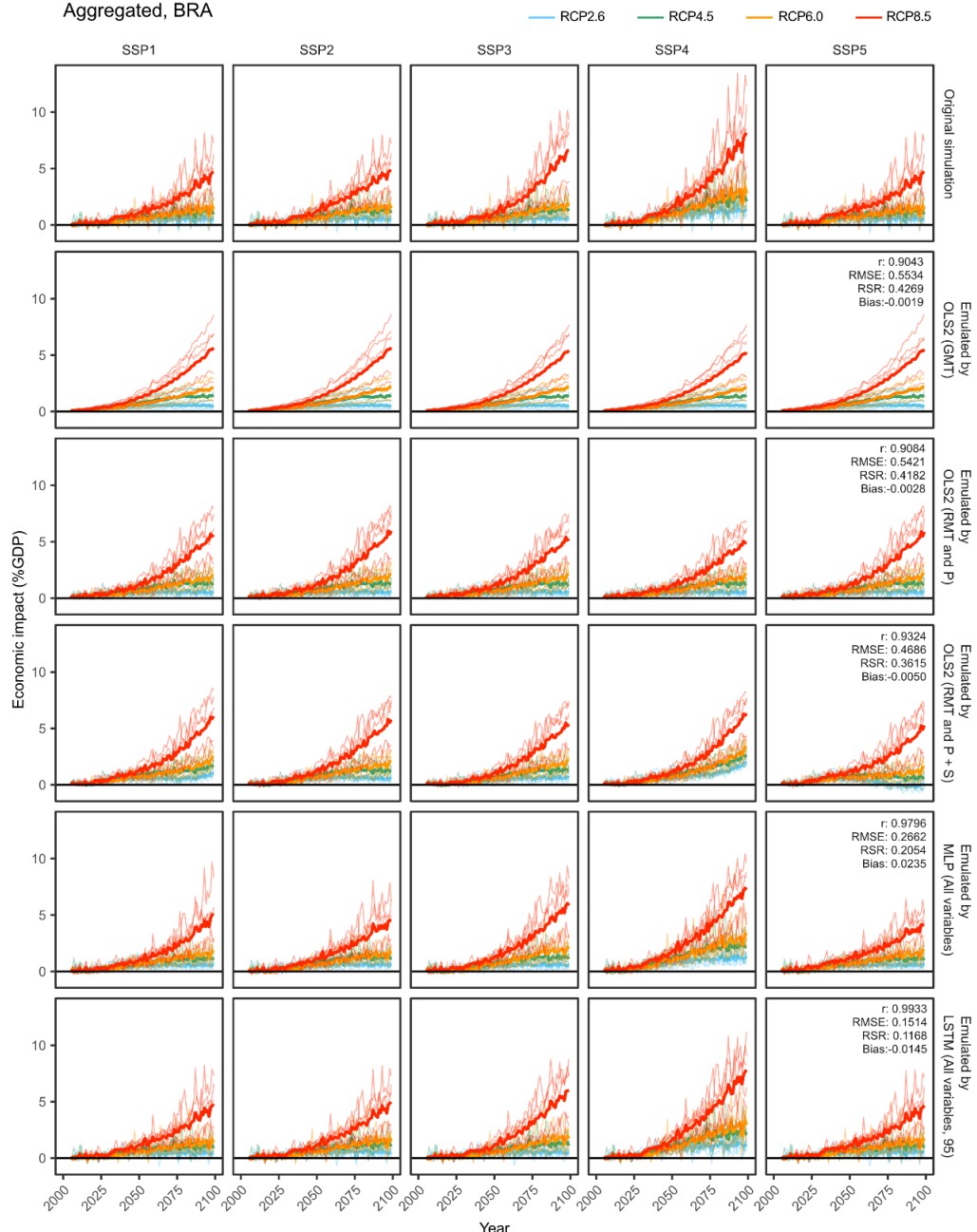

**Figure 8: Time-series results of the simulation and emulations for aggregated economic impacts in Brazil region.**
OLS2 (GMT): OLS2 with global mean temperature. OLS2 (RMT and P): OLS2 with regional mean temperature and precipitation. OLS2 (RMT and P +S): OLS2 with regional mean temperature precipitation and socioeconomic variables. MLP (All variables): MLP with all the input variables. LSTM (All variables, 95): LSTM with all the input variables for 95 years. Thin lines represent individual GCM results and bold lines represent average of 5 GCMs.

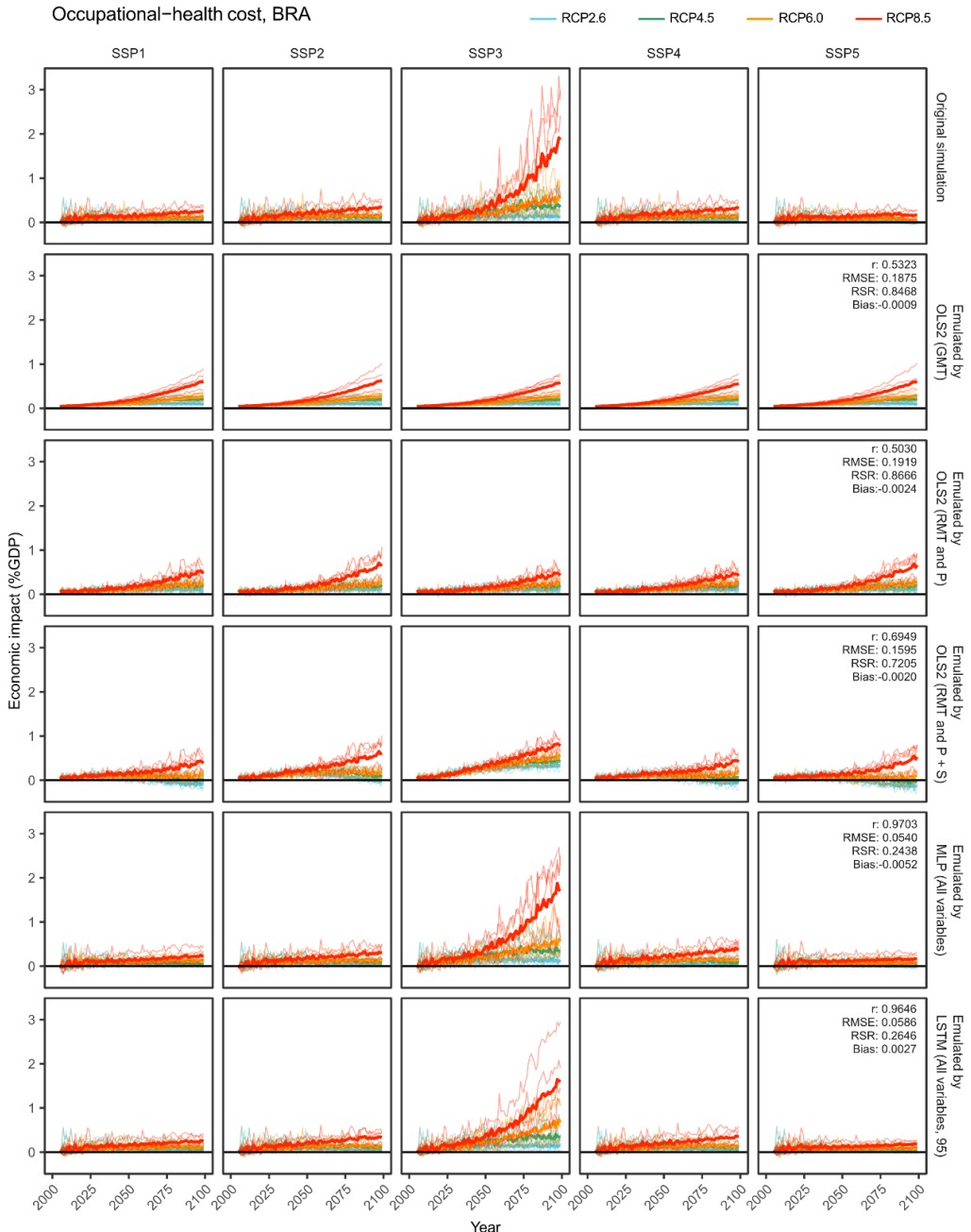

**Figure 9: Time-series results of the simulation and emulations for the occupational-health cost sector in Brazil region.** OLS2 (GMT): OLS2 with global mean temperature. OLS2 (RMT and P): OLS2 with regional mean temperature and precipitation. OLS2 (RMT and P +S): OLS2 with regional mean temperature precipitation and socioeconomic variables. MLP (All variables): MLP with all the input variables. LSTM (All variables, 95): LSTM with all the input variables for 95 years. Thin lines represent individual GCM results and bold lines represent average of 5 GCMs.

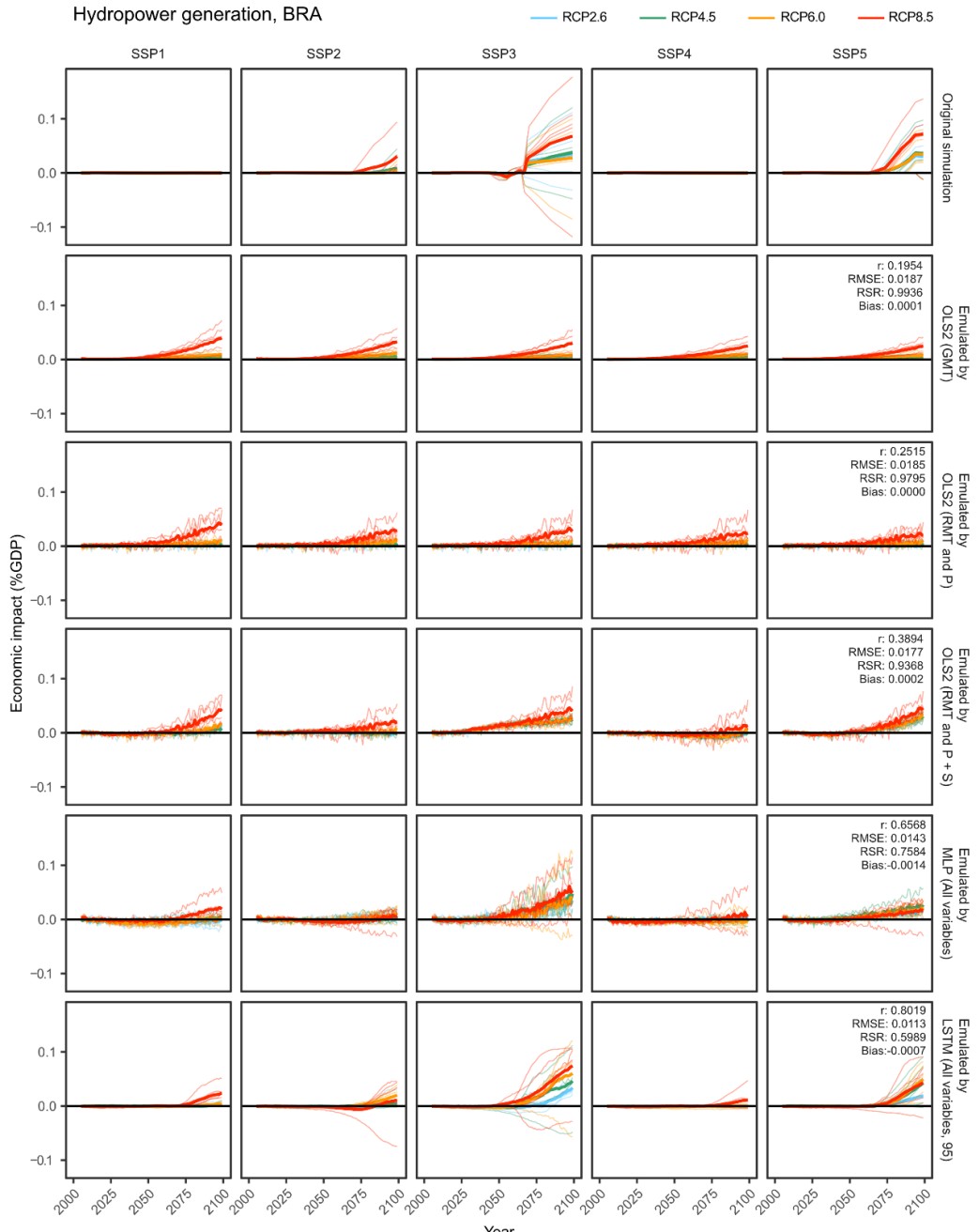

**Figure 10: Time-series results of the simulation and emulations for the hydropower generation sector in Brazil region. OLS2 (GMT): OLS2 with global mean temperature. OLS2 (RMT and P): OLS2 with regional mean temperature and precipitation. OLS2 (RMT and P +S): OLS2 with regional mean temperature precipitation and socioeconomic variables. MLP (All variables): MLP with all the input variables. LSTM (All variables, 95): LSTM with all the input variables for 95 years. Thin lines represent individual GCM results and bold lines represent average of 5 GCMs**

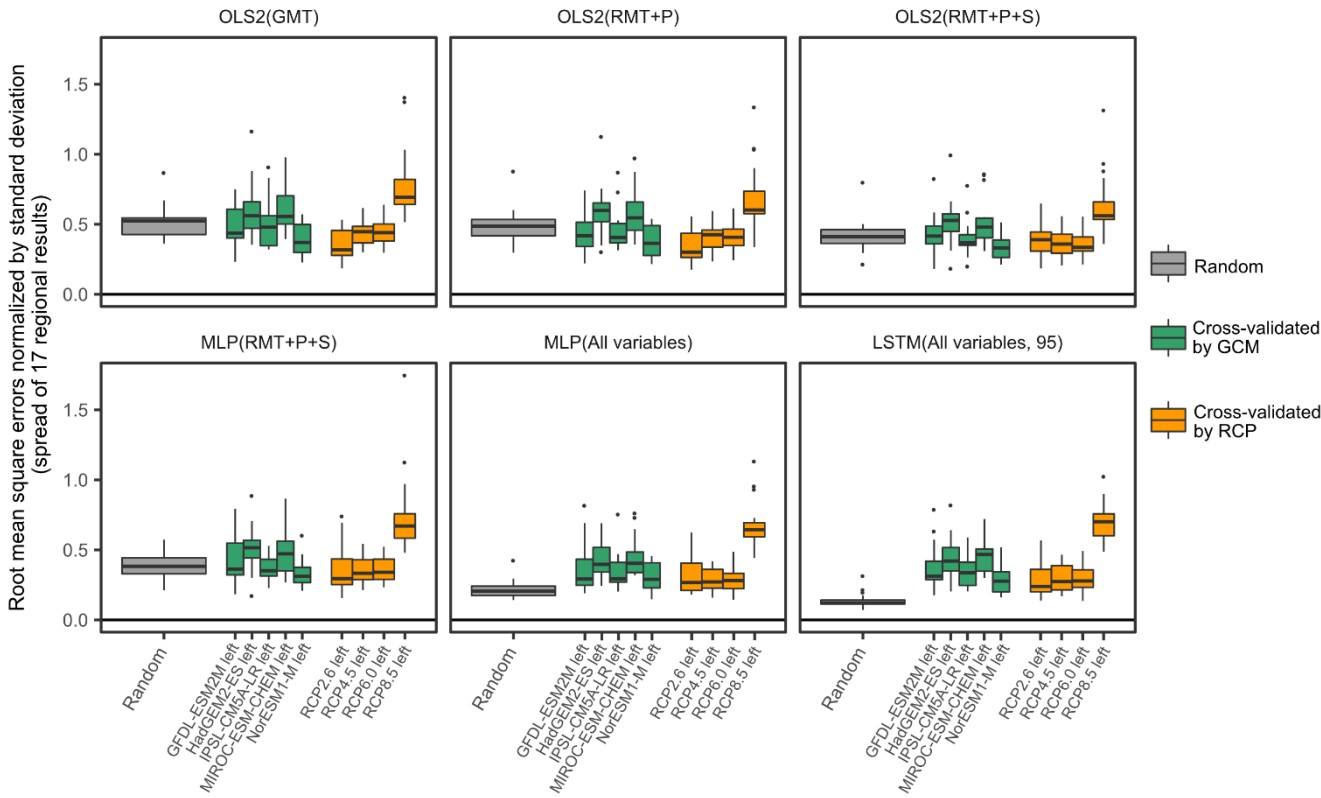

**Figure 11: Performance of emulations of the aggregated impacts under different cross-validation procedures. Root mean squared errors (normalized by pooled standard deviation) between simulation results and emulation results are shown. Unlike the correlation coefficient, a higher value means a larger error. Random means training and test scenarios are selected randomly. For example, 'GFDL-ESM2M left' denotes that the emulators are trained by the results of HadGEM2-ES, IPSL-CM5A-LR, MIROC-ESM-CHEM, and NorESM1-M, then tested by the results of GFDL-ESM2M. Similarly, for example, 'RCP2.6 left' denotes that the emulators are trained by the results of RCP4.5, RCP6.0, and RCP8.5, then tested by the results of RCP2.6. OLS2 (GMT): OLS2 with global mean temperature. OLS2 (RMT+P): OLS2 with regional mean temperature and precipitation. OLS2 (RMT+P+S): OLS2 with regional mean temperature precipitation and socioeconomic variables. MLP (RMT+P+S): MLP with regional mean temperature precipitation and socioeconomic variables. MLP (All variables): MLP with all the input variables. LSTM (All variables, 95): LSTM with all the input variables for 95 years.**