# Peer review of "Reproducing complex simulations of economic impacts of climate change with lower-cost emulators"

_Geoscientific Model Development, 2020_

## Referee Comment (RC1) · Anonymous Referee #1 · 11 Feb 2021

Summary: The authors present a hierarchy of emulators, which progressively take in more comprehensive input data and have more complicated internals. They show that even linear or quadratic functions of global mean temperature (OLS1/OLS2) provide reasonable estimates of aggregated economic damages, and attribute this to its straight-forward relationship with heat-induced mortality, which dominates the aggregate. For the economic impacts in more complicated sectors, however, more sophisticated emulators perform better. While the present method seems useful, it is not clear to me how the authors envision others will use it– and I do not think they have adequately made their case why others should.

[Figure]

General Comments: This manuscript was a pleasure to read and thoroughly interesting- I commend the authors on their great work! My only comment is that the authors provide no direct evidence of "reducing the implementation and computational costs" of impact calculations (this may be obvious to some expert readers, but not all).

On implementation cost: While running the authors' emulators requires no sector-specific knowledge, interpreting the emulators' results does require domain-specific knowledge (as evidence by some of the nuanced discussions of the limitations of the method in this manuscript). Similarly, many users might wish to re-train the emulators based on different GCMs or IAMs, which again would likely require domain-specific knowledge.

On computational cost: The authors claim, without much evidence, that the computational cost of the emulators is overall much less than the computational cost of the base simulations. They mostly attribute this to the computational cost of the geophysical simulations, and mention that the economic sector models are typically more inexpensive-they even propose a hybrid approach where geophysical impacts are emulators then fed into sector-specific economic simulations. However, there are no specifics about the compute time or memory constraints of any of the methods.

On line 365, the authors state: "While computational cost of emulation is small in the calculation (prediction) phase, even by the most complex emulator used in this study (on the order of milliseconds), the availability of input variables in context-specific. For example, the cost of preparing or generating sub-yearly regional climate variables should also be considered".

I understand why the authors focus on the "prediction" phase, but it would be useful to know more specific about the computational cost of the "training" phase of each emulator, as well as the "preprocessing" phase. In particular, I would recommend the authors make a table of the required for each of these three phases: storage space (e.g. for raw input data) and CPU/GPU configuration and runtime. While I agree with the authors

that using the pre-trained emulators presented by the authors has a negligible computational cost compared to running a full GCM/IAM simulation, it is not clear to me that the entire process of developing-training-running an emulator is more inexpensive.

Is the primary product of this paper a pre-trained "out-of-the-box" emulator that the authors want authors to use? Or is it the method that other authors can follow to develop their own? This should be clarified in the text.

Specific Comments: - I appreciate the authors sharing their code publicly, but would recommend they at least include a README file with instructions on how to run the emulators. It was not obvious to me how I would replicate their analysis based on the files in the Zenodo directory. If the authors want their emulator to be widely used as an alternative to IAMs, I would recommend publishing it with documentation in a version-controlled and public-facing repository, e.g. on Github/Gitlab.

- line 250: the authors should mention here that the performance for aggregated impacts is only good because– based on Supplemental Figure 3 of their 2019 paper– they are dominated (>90%) by heat-related excess mortaility and occupational health costs, which themselves have high performance when just the global-mean temperature is used. This context is quite important and may not be obvious to readers. This is mentioned on lines 299-301 of the discussion, but even then referring to these are the "main contributors" feels like an understatement, given that they represent about 90% of the impacts in almost all scenarios.

- line 329-331: How do we know that such overfitting or "leakage" does not affect other sectors? Should we not also be cautious about other results of the model?

---

## Referee Comment (RC2) · Anonymous Referee #2 · 19 Feb 2021

This paper presents a series of strategies for emulating economic impacts of climate change, progressively adding more explanatory variables and using more complex emulation techniques. The authors consider nine impact sectors under a range of scenarios, considering 5 socioeconomic pathways, 4 climate pathways and 5 climate models (to capture modelling uncertainties). The underlying impacts calculations are complex and cannot easily be performed by non-specialists, therefore justifying the need for user-friendly emulators. The paper was a pleasure to read.

What is the main objective of the paper? Is it to provide a tool for others to use, to provide a methodology for others to apply to their own data, or to use emulation to

extract understanding of the underlying models? The paper addresses all of these, and abstract presents the main objective as the first "The developed emulators could be used to explore future scenarios related to climate-change policies", but it is not clear how the emulators can be applied at present. What are the likely applications of the emulators, to apply them to another data set, most likely using a different climate scenario (or climate model)? It would be very useful, and appropriate for GMD, if the authors provide documented code in a form suitable for this. For instance an application directory which contains the trained emulators (or code to generate them) and an example input data set, together with detailed instructions how to run the code and construct the input data set.

Relevant to the above, the cross-validation selects input scenarios at random from the 100 combinations (SSP*RCP*model). This is not an especially strong test as the training data will always include some instances of each of the SSPs, RCPs and climate models. A stronger test would be to test under specific LOO assumptions. i.e. How well are the impacts under each RCP estimated from an emulator built only with the other RCPs? How well are the impacts using each climate model estimated by an emulator built only with other climate models? These analyses would give confidence of the applicability to independent climate data, which seems to be where the real power of this approach lies.

The authors use a simple regression fitting. Did they consider building in a stepwise fashion using e.g. and AIC criteria (e.g. stepAIC function in R)? This can significantly reduce over-fitting, especially when there are many inputs, and improve performance under cross-validation. The authors should consider looking at this if, as I suspect, it is straightforward to implement. It may be as simple as adding a line of code e.g.

require(MASS)

model <- lm(output ∼ v1+v2+v3+v4+. . . etc) #what you have already?

model <- stepAIC(model) #remove terms that don't satisfy AIC

I was not convinced by the correlation between model fit and impact magnitude (Figure 7 etc). I would like to see these data points labelled by sector. For instance, the largest impacts are heat-related deaths and occupational health. These are both temperature-driven impacts and I would expect them to be easier to emulate because precipitation is more difficult to model and with a more complex spatiotemporal response. Conversely, the most difficult impacts to emulate are fluvial floods and hydropower, which are likely sensitive to the details of the precipitation projections. Related to this, I do not regard the statement that aggregate impacts are easier to emulate as being robust. I suspect this result is a function of the data set, reflecting the fact that the largest impacts are (happen to be?) in those (temperature-dependent) sectors which are the easiest to emulate?

Line 59 should mention Gaussian Process emulators as an alternative to ANN. At least some of the cited references used GPs. This is a widely used emulation approach and has a number of advantages, most notably by estimating the uncertainty of emulated predictions.

---

## Author Comment (AC1) · 16 Apr 2021

Thank you for the valuable comments which have helped us to increase the quality and clarity of the manuscript. As requested, we have conducted additional analyses and revised the manuscript accordingly. We also modified the disclosed code and data based on suggestions by the referees. Please see the supplement PDF file for point-by-point responses.

–

Please also note the supplement to this comment:
https://gmd.copernicus.org/preprints/gmd-2020-349/gmd-2020-349-AC1-supplement.pdf

---

## Author Response (AR1)

Dear Editors and Referees

Thank you for the valuable comments which have helped us to increase the quality and clarity of the manuscript. As requested, we have conducted additional analyses and revised the manuscript accordingly. We also modified the disclosed code and data based on suggestions by the referees. We give point-by-point responses to the comments below. Line numbers referred to in the responses pertain to the revised manuscript in track-changes.

**Referee #1**

**Referee comment:**

This manuscript was a pleasure to read and thoroughly interesting- I commend the authors on their great work! My only comment is that the authors provide no direct evidence of "reducing the implementation and computational costs" of impact calculations (this may be obvious to some expert readers, but not all).

**Our response:**

Thank you for reviewing our manuscript and pointing out the lack of evidence concerning the reduction in computational cost. We agree that it was not obvious to all potential readers of this article. Accordingly, we have added information on the computational cost in the revised manuscript. Please also see the responses below.

**Referee comment:**

On implementation cost: While running the authors' emulators requires no sector specific knowledge, interpreting the emulators' results does require domain-specific knowledge (as evidence by some of the nuanced discussions of the limitations of the method in this manuscript). Similarly, many users might wish to re-train the emulators based on different GCMs or IAMs, which again would likely require domain-specific knowledge.

**Our response:**

As the referee points out, there are two main groups of potential readers of this article. The first group wish to use the emulators which have been developed in this study, and the second group wish to develop their own emulators using their data. For the second group, domain-specific knowledge is required to better design the emulators as discussed in the manuscript. Even for the first group, domain-specific knowledge is useful so as not to misinterpret the results. We agree that this was not clear in the manuscript and could be misleading, so we modified the relevant sentence (Line 442-443). In addition, as well as domain-specific knowledge, knowledge

on statistical models is also needed so as not to misuse and misinterpret the results. Thus, we have added a paragraph explaining several points that users should be aware of (Lines 447-468).

**Referee comment:**

On computational cost: The authors claim, without much evidence, that the computational cost of the emulators is overall much less than the computational cost of the base simulations. They mostly attribute this to the computational cost of the geophysical simulations, and mention that the economic sector models are typically more inexpensive they even propose a hybrid approach where geophysical impacts are emulators then fed into sector-specific economic simulations. However, there are no specifics about the compute time or memory constraints of any of the methods.

On line 365, the authors state: "While computational cost of emulation is small in the calculation (prediction) phase, even by the most complex emulator used in this study (on the order of milliseconds), the availability of input variables in context-specific. For example, the cost of preparing or generating sub-yearly regional climate variables should also be considered".
I understand why the authors focus on the "prediction" phase, but it would be useful to know more specific about the computational cost of the "training" phase of each emulator, as well as the "preprocessing" phase. In particular, I would recommend the authors make a table of the required for each of these three phases: storage space (e.g. for raw input data) and CPU/GPU configuration and runtime. While I agree with the authors that using the pre-trained emulators presented by the authors has a negligible computational cost compared to running a full GCM/IAM simulation, it is not clear to me that the entire process of developing-training-running an emulator is more inexpensive.

**Our response:**

For example, concerning the simulation of the economic impact of changes in hydropower generation capacity, one scenario (100 years) requires around 15 to 20 hours for bio/physical impact simulation and around 1.5 hours for general-equilibrium-based economic simulation. We have added this information in the revised manuscript (Lines 414-415).

We have also added an explicit assessment of computational cost (Lines 277-281). The results are shown in Table 7. In summary, the computational cost of running emulators is small, while developing and training emulators is non-negligible. This information is important to demonstrate the usefulness of the developed method.

**Referee comment:**

Is the primary product of this paper a pre-trained "out-of-the-box" emulator that the authors want authors to use? Or is it the method that other authors can follow to develop their own? This should be clarified in the text.

**Our response:**

We fully agree that this was not clear in the previous version of the manuscript. As mentioned in the above response, we think there are two main groups of potential readers of this article, i.e., users and developers of emulators. However, the code/data associated with the previous manuscript were not 'user-friendly', particularly for the former group.

We decided to separate the disclosed source code/data into two directories. The first directory is for reproducing the analysis conducted in this article, and the second directory is for users of the pre-trained emulators. In the revised manuscript, we have also added a paragraph on our expectations concerning the readers (Lines 100-104), and we added another paragraph particularly for users to read (Lines 447-468).
* * *
**Referee comment:**

I appreciate the authors sharing their code publicly, but would recommend they at least include a README file with instructions on how to run the emulators. It was not obvious to me how I would replicate their analysis based on the files in the Zenodo directory. If the authors want their emulator to be widely used as an alternative to IAMs, I would recommend publishing it with documentation in a version-controlled and public-facing repository, e.g. on Github/Gitlab.

**Our response:**

As mentioned above, we have separated the disclosed data into two parts, and added README files. One obstacle to using GitHub/GitLab is the limitation of the file size. Relatively large files are required to reproduce our analysis, and that's why we choose Zenodo. Version control is also possible in Zenodo with DOI (https://blog.zenodo.org/2017/05/30/doi-versioning-launched/), and thus we think Zenodo is an appropriate platform to disclose the code and data. However, we agree that GitHub/GitLab is suitable particularly for collaborative development. If this research moves in a more collaborative direction in future, we will certainly consider using GitHub/GitLab.

**Referee comment:**

line 250: the authors should mention here that the performance for aggregated impacts is only good because– based on Supplemental Figure 3 of their 2019 paper– they are dominated (>90%) by heat-related excess mortaility and occupational health costs, which themselves have high performance when just the global-mean temperature is used. This context is quite important and may not be obvious to readers. This is mentioned on lines 299-301 of the discussion, but even then referring to these are the "main contributors" feels like an understatement, given that they represent about 90% of the impacts in almost all scenarios.

**Our response:**

To more accurately express the contribution of these two sectors, we changed the word "main" to "dominant" (Line 359). We have also added a figure which indicates the sectoral contributions to the aggregated impact in the SI.
* * *
**Referee comment:**

- line 329-331: How do we know that such overfitting or "leakage" does not affect other sectors? Should we not also be cautious about other results of the model?

**Our response:**

In general, unfortunately, leakage is difficult to detect because it can pass the cross-validation test, unlike simple overfitting. Proving the absence of leakage is quite difficult ("Testing shows the presence, not the absence of bugs" as Dijkstra said) and ad-hoc/heuristic approaches are needed to avoid leakage. While there is no perfect solution, it can be effective to think about the actual situation in which the developed emulators will be used. For example, if the emulators will be used to estimate economic impacts under different RCPs (emission pathways), which are not included in the training data, cross-validation by RCPs can be effective to estimate the actual performance of the emulators in that situation. This is related to a comment from Referee 2, and we have added additional analysis (shown in Fig. 11 and Lines 338-348) and discussion (Lines 448-455) in the revised manuscript.

**Referee #2**

**Referee comment:**

This paper presents a series of strategies for emulating economic impacts of climate change, progressively adding more explanatory variables and using more complex emulation techniques. The authors consider nine impact sectors under a range of scenarios, considering 5 socioeconomic pathways, 4 climate pathways and 5 climate models (to capture modelling uncertainties). The underlying impacts calculations are complex and cannot easily be performed by non-specialists, therefore justifying the need for user-friendly emulators. The paper was a pleasure to read.

What is the main objective of the paper? Is it to provide a tool for others to use, to provide a methodology for others to apply to their own data, or to use emulation to extract understanding of the underlying models? The paper addresses all of these, and abstract presents the main objective as the first "The developed emulators could be used to explore future scenarios related to climate-change policies", but it is not clear how the emulators can be applied at present. What are the likely applications of the emulators, to apply them to another data set, most likely using a different climate scenario (or climate model)? It would be very useful, and appropriate for GMD, if the authors provide documented code in a form suitable for this. For instance an application directory which contains the trained emulators (or code to generate them) and an example input data set, together with detailed instructions how to run the code and construct the input data set.

**Our response:**

Thank you for reviewing our manuscript. We agree that the objective of this paper was not clear. This is also related to a comment from Referee 1. We expect that there are two main groups of potential readers of this article. The first group wish to use the emulators which have been developed in this study, and the second group wish to develop their own emulators using their data. We have added a paragraph concerning our expectations about the readers (Lines 100-104).

We expect there are several situations in which the developed emulators can be used. Applying the emulators to a different climate scenario is a typical one, but this is not the only application. We feel this point was too briefly discussed in the previous manuscript, and we have therefore elaborated the description on how the emulators can be used (Lines 47-62).

As the referee comments, providing ready-to-use code and data is useful for the users of the developed emulators, and thus we decided to separate disclosed code/data into two directories. One is for model users, and the other is for reproducing the analyses conducted in this article.

**Referee comment:**

Relevant to the above, the cross-validation selects input scenarios at random from the 100 combinations (SSP*RCP*model). This is not an especially strong test as the training data will always include some instances of each of the SSPs, RCPs and climate models. A stronger test would be to test under specific LOO assumptions. i.e. How well are the impacts under each RCP estimated from an emulator built only with the other RCPs? How well are the impacts using each climate model estimated by an emulator built only with other climate models? These analyses would give confidence of the applicability to independent climate data, which seems to be where the real power of this approach lies.

**Our response:**

To explore the performance of the emulators under such situations, we conducted additional analysis by changing how training and test data are chosen: leave-one-GCM-out and leave-one-RCP-out cross-validation (Line 257-262). The results show that if the situation corresponds to extrapolation, for example, the emulators are trained by RCP2.6, 4,5, and 6.0 data and then applied to RCP8.5 data, then the errors tend to be larger. We added these results (Fig. 11, and Lines 338-348) and a discussion on this point (Lines 448-455) in the revised manuscript. We think these results are important for interpreting the characteristics and actual performance of the emulators.
* * *
**Referee comment:**

The authors use a simple regression fitting. Did they consider building in a stepwise fashion using e.g. and AIC criteria (e.g. stepAIC function in R)? This can significantly reduce over-fitting, especially when there are many inputs, and improve performance under cross-validation. The authors should consider looking at this if, as I suspect, it is straightforward to implement. It may be as simple as adding a line of code e.g.

require(MASS)

model <- lm(output ~ v1+v2+v3+v4+: : : etc) #what you have already?

model <- stepAIC(model) #remove terms that don't satisfy AIC

**Our response:**

We applied the stepAIC function to the lm models with many input variables as the referee suggested. However, variable selection had a negligible effect on the emulator performance. This implies that overlearning is not the main cause of the performance degradation, at least for the tested OLS-based models. We mention variable selection in the discussion (Line 424-428) and in the SI.

**Referee comment:**

I was not convinced by the correlation between model fit and impact magnitude (Figure 7 etc). I would like to see these data points labelled by sector. For instance, the largest impacts are heat-related deaths and occupational health. These are both temperature driven impacts and I would expect them to be easier to emulate because precipitation is more difficult to model and with a more complex spatiotemporal response. Conversely, the most difficult impacts to emulate are fluvial floods and hydropower, which are likely sensitive to the details of the precipitation projections. Related to this, I do not regard the statement that aggregate impacts are easier to emulate as being robust. I suspect this result is a function of the data set, reflecting the fact that the largest impacts are (happen to be?) in those (temperature-dependent) sectors which are the easiest to emulate?

**Our response:**

We have added sector labels in Fig. 7 as the referee suggests. While we showed the correlation between the magnitude of the impact and the performance of emulations, we did not intend to claim the existence of causality (i.e., higher magnitude in the impacts caused higher performance of the emulation or vise versa). Also, we did not intend to claim the actual aggregated impacts are easier to emulate, but the simulated aggregated impacts used in this study. For clarity, we have modified the relevant sentences (Lines 364-365, and Line 368).

**Referee comment:**

Line 59 should mention Gaussian Process emulators as an alternative to ANN. At least some of the cited references used GPs. This is a widely used emulation approach and has a number of advantages, most notably by estimating the uncertainty of emulated predictions.

**Our response:**

We mention Gaussian Process emulators (Lines 430-431) and other alternative models and techniques (Lines 429-430) in the revised paper.